# Adaptive Temperature Enhanced Dual-level Hypergraph Contrastive Learning

## Abstract

Hypergraphs, which incorporate hyperedges to link multiple nodes and capture complex high-order relationships, have attracted increasing attention in recent years. Consequently, a bunch of hypergraph neural networks has been proposed to model the high-order relationships between hyperedges and nodes. Inspired by the success of graph contrastive learning, researchers have begun exploring the benefits of contrastive learning over hypergraphs. However, these works still have the following limitations in modeling the high-order relationships over unlabeled data: (i) They primarily focus on maximizing the agreements among individual node embeddings while neglecting the capture of group-wise collective behaviors within hypergraphs; (ii) Most of them disregard the importance of the temperature index in discriminating contrastive pairs during contrast optimization. To address these limitations, we propose a novel **Ad**aptive **T**emperature enhanced **Hy**per**G**raph **C**ontrastive **L**earning framework called **AdT-HyGCL** to boost contrastive learning over hypergraphs. Specifically, we first introduce a noise enhancement module to generate relatively challenging augmented hypergraphs for hypergraph contrastive tasks. Unlike most works that merely maximize the agreement of node embeddings in hypergraphs, we then propose a dual-level contrast mechanism that not only captures the individual node behaviors in a local context but also models the group-wise collective behaviors of nodes within hyperedges from a community perspective. Furthermore, we design an adaptive temperature-enhanced contrastive optimization to improve the discrimination ability between positive and negative contrastive pairs, thereby facilitating more effective hypergraph representation learning. Theoretical justifications and empirical experiments conducted on eight benchmark hypergraphs demonstrate that AdT-HyGCL exhibits excellent rationality, generalization, effectiveness, and robustness compared to state-of-the-art baseline models.

## 1 Introduction

Graphs are widely used to represent pairwise interactions between entities. However, they have limitations in capturing intricate relationships and higher-order group-wise structures. In contrast, hypergraphs provide a more versatile structure by introducing hyperedges that can connect multiple nodes to represent complex relationships. They have been explored in various domains, including social networks, knowledge graphs, biological networks, recommend systems, and transportation networks (Li et al., 2013; An et al., 2021; Ma et al., 2022; Zhang et al., 2022; Xia et al., 2022). To leverage the benefits of hypergraphs, a bunch of hypergraph neural networks (HyGNNs) (Feng et al., 2019; Dong et al., 2020; Bai et al., 2021) have been developed to model the rich connectivity patterns within hypergraphs in supervised or semi-supervised settings. Inspired by the success of graph contrastive learning, recent works (Wei et al., 2022; Cai et al., 2022) extend contrastive learning to hypergraphs for modeling hypergraph structures via HyGNNs over unlabeled data.

However, current contrastive learning methods over hypergraphs (Wei et al., 2022; Cai et al., 2022) still have limitations in modeling the high-order relationships and collective behaviors within hypergraphs over unlabeled data: (i) These works primarily focus on maximizing agreements among node embeddings while neglecting the capture of group-wise behaviors within hypergraphs; (ii) They consider the temperature index in contrastive learning as a hyper-parameter while underestimating the importance of temperature in differentiating contrastive pairs during contrast optimization.

To handle the above challenges, we design a novel **Ad**aptive **T**emperature enhanced dual-level **Hy**per**G**raph **C**ontrastive **L**earning framework called **AdT-HyGCL** to enhance the contrastive learning over hypergraphs. Specifically, we first introduce a noise-enhanced module over augmented hypergraphs to generate challenging hypergraph pairs. Afterward, the noise-enhanced hypergraph augmentations are fed to the HyGNNs encoder to obtain the node embeddings and the hyperedge embeddings. To address the first challenge, we design a dual-level contrast mechanism that aims to maximize the agreements among individual node embeddings at the node level and also focuses on capturing the group-wise behaviors within hyperedges at the community level, simultaneously. To handle the second challenge, we first prove the importance of temperature in hypergraph contrastive learning and further design an adaptive temperature-enhanced contrast optimization to dynamically adjust the temperature during dual-level contrast optimization. This adaptation serves to enhance the discriminative capacity between contrastive pairs. By the aforementioned steps, AdT-HyGCL can comprehensively model the high-order relationships in hypergraphs over unlabeled data and enhance hypergraph contrastive learning. To conclude, this work makes the following contributions:

- *Novelty:* We design a novel hypergraph contrastive learning framework integrating the dual-level contrast strategy and the adaptive temperature-enhanced contrast optimization to pre-train HyGNNs encoder over unlabeled data.
- *Generalization:* AdT-HyGCL is designed as a general framework that unifies various hypergraph augmentations and different contrast optimizations to boost hypergraph representation learning and further enhance the model performance over downstream tasks.
- *Effectiveness and Robustness:* Theoretical justifications and empirical experiments over eight benchmark hypergraphs demonstrate the rationality, effectiveness, and robustness of AdT-HyGCL.

## 2 RELATED WORK

**Hypergraph Neural Networks.** Hypergraph neural networks (HyGNNs) (Li et al., 2018; Zhang et al., 2020; Feng et al., 2019; Cheng et al., 2022) have gained significant attention in recent years due to their ability to capture complex relationships among nodes in hypergraphs. One of the notable works is Hypergraph Neural Network (HGNN) (Feng et al., 2019), which designs a hyperedge convolution operator to formulate complex and high-order data through its hypergraph structure. Another notable work, AllDeepSets (Chien et al., 2022), leveraging the deep multiset functions to propagate and aggregate the information among hypergraphs, has gained excellent performance on various benchmark datasets. Motivated by existing HyGNNs, this work proposes to design a hypergraph contrastive learning framework to learn the complex relationships over unlabeled data.

**Hypergraph Contrastive Learning.** Existing graph contrastive learning (GCL) models (Wang et al., 2021; Yu et al., 2022; Zhu et al., 2021b; Tong et al., 2021; Trivedi et al., 2022) leverage different types of data transformations to augment graphs into different views and further train the encoder by discriminating positive pairs and negative pairs generated from unlabeled data. Inspired by these works, researchers start to explore the benefits of contrastive learning over hypergraphs (Wei et al., 2022; Cai et al., 2022; Song et al., 2023; Lee & Shin, 2023). For instance, HyperGCL (Wei et al., 2022) proposes a generative method to create generative augmentations of hypergraphs. CHGNN (Song et al., 2023) designs an adaptive augmentation strategy for hypergraph augmentation and further proposes the updated hypergraph encoder to learn the node embedding over the unlabeled data. However, these works still have limitations in describing the group-wise collective node behaviors within hyperedges during hypergraph contrastive learning. To handle this, TriCL (Lee & Shin, 2023) proposes to maximize the mutual information between nodes, hyperedges, and groups in the embedding space. However, TriCL still fails to comprehensively depict the group-wise collective behaviors within hyperedges. Besides, they do not consider the influence of temperature index in contrastive optimization. Motivated by these works, we propose to design a temperature-enhanced dual-level hypergraph contrastive learning framework to reach agreements among individual node embeddings and community embeddings.

## 3 PRELIMINARY

**Hypergraph Neural Networks.** Given a hypergraph $\mathcal{G} = (\mathcal{V}, \mathcal{E}, \mathcal{X})$, where $\mathcal{V}$ is the set of nodes, $\mathcal{E}$ is the set of hyperedges, $\mathcal{X}$ is the attribute features set of nodes and hyperedges. Unlike the pair-

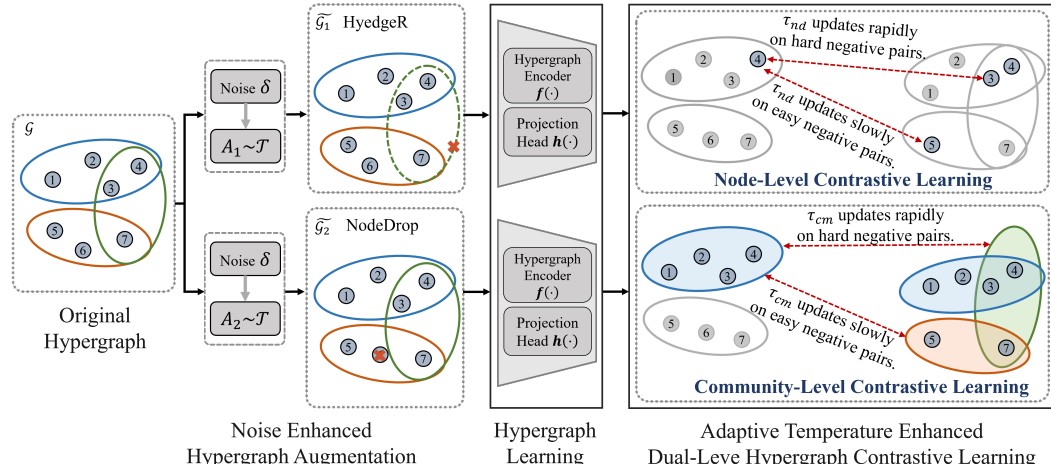

Figure 1: The framework of AdT-HyGCL: (i) given a hypergraph $\mathcal{G}$, AdT-HyGCL samples $A_1, A_2$ from the hypergraph augmentation set $\mathcal{T}$. Here $A_1$ and $A_2$ denote hyperedge removal and node dropping, respectively. With the augmented hypergraphs via $A_1$ and $A_2$, it performs the noise $\delta$ over augmented graphs for generating challenging hypergraph pairs; (ii) the augmented graph $\widetilde{G}_1$ and $\widetilde{G}_2$ are fed into HyGNNs encoder $f(\cdot)$ and projection head $h(\cdot)$ to get the node and hyperedge embeddings. (iii) a dual-level contrastive strategy is designed to reach agreements among node embeddings from a local view and agreements among community embeddings from a global perspective. The dual-level contrast optimization is enhanced via the adaptive temperature $\tau_{nd}$ and $\tau_{cm}$, respectively.

wise edge in graphs that only connects two nodes, a hyperedge $e \in \mathcal{E}$ can link any number of nodes. A hypergraph can be represented by the incidence matrix $H$, where $H_{ve} = 1$ if $v \in e$. Otherwise, $H_{ve} = 0$. For each node $v \in \mathcal{V}$ and hyperedge $e \in \mathcal{E}$, we leverage $d(v) = \sum_{e \in \mathcal{E}} H_{ve}$ and $d(e) = \sum_{v \in \mathcal{V}} H_{ve}$ to denote the node degree and hyperedge degree, respectively. There are a bunch of works about HyGNNs (Dong et al., 2020; Feng et al., 2019; Bai et al., 2021; Yadati et al., 2019) designed to map the hypergraph to a $b$-dimension latent representation space via function $f : \mathcal{G} \rightarrow \mathbb{R}^b$ with higher-order message passing among nodes and hyperedges. In this paper, we choose AllDeepSets (Chien et al., 2022) as the hypergraph encoder to learn the node embeddings and the hyperedge embeddings, which can be formulated as follows:

$$\mathbf{Z}_{e,:}^{(t+1)} = f_{\mathcal{V} \rightarrow \mathcal{E}}(V_e, \mathbf{U}^{(t)}; \mathbf{Z}_{e,:}^{(t)}), \quad \mathbf{U}_{v,:}^{(t+1)} = f_{\mathcal{E} \rightarrow \mathcal{V}}(E_v, \mathbf{Z}^{(t+1)}; \mathbf{U}_{v,:}^{(t)}), \tag{1}$$

where $\mathbf{Z}_{e,:}^{(t+1)}$ is the embedding of hyperedge $e$ at time $t + 1$, $\mathbf{U}_{v,:}^{(t+1)}$ is the embedding of node $v$, $f_{\mathcal{V} \rightarrow \mathcal{E}}$ and $f_{\mathcal{E} \rightarrow \mathcal{V}}$ are two multiset functions in terms of the input. $\mathbf{Z}_{e,:}^0$ denotes the original attribute feature of hyperedge $e$ and $\mathbf{U}_{v,:}^0$ denotes the attribute feature of node $v$, where $\mathbf{U}_{v,:}^0 = \mathbf{X}_{v,:}$.

**Graph Contrastive Learning.** The main idea of graph contrastive learning aims to maximize the agreements among positive and negative contrastive pairs over unlabeled data. This work follows SimCLR (Chen et al., 2020) framework to conduct contrastive learning. Specifically, given a graph $\mathcal{G}$, SimCLR first obtains the augmented graph pairs $\mathcal{G}_1, \mathcal{G}_2$ via graph augmentation methods. Then the augmented graph pairs are further processed by graph neural networks (GNNs) backbone $f$, outputting node embeddings $\mathbf{U}^1 = f(\mathbf{X}_1, \mathbf{A}_1)$ and $\mathbf{U}^2 = f(\mathbf{X}_2, \mathbf{A}_2)$, where $\mathbf{X}_*$ and $\mathbf{A}_*$ denote the attribute features and adjacent matrices of the corresponding augmented graph $\mathcal{G}_*$. After passing through a nonlinear projection head $h(\cdot)$, the transformed embedding pair set $\mathbf{U}^1, \mathbf{U}^2$ are optimized under the following NT-Xent loss $l_{NT}$:

$$\mathcal{L}_{\text{NT}} = -\log \sum_{\substack{v_i \in \mathcal{V}}} \frac{\exp(\text{sim}(\mathbf{u}_i^1, \mathbf{u}_i^2)/\tau)}{\sum\limits_{k \neq i} \exp(\text{sim}(\mathbf{u}_i^1, \mathbf{u}_k^2)/\tau) + \exp(\text{sim}(\mathbf{u}_i^1, \mathbf{u}_i^2)/\tau)}, \tag{2}$$

where $\tau$ is the temperature index to control the sharpness of the probability distribution.

**Problem 1.** *Hypergraph Contrastive Learning. Given a hypergraph $\mathcal{G} = (\mathcal{V}, \mathcal{E}, \mathcal{X})$, we aim to design a comprehensive hypergraph contrastive learning framework to train hypergraph encoder $f(\cdot)$ over unlabeled data and further fine-tune the encoder over labeled data for downstream tasks.*

## 4 METHODOLOGY

In this section, we present the details of AdT-HyGCL which includes three modules: (i) noise-enhanced hypergraph augmentation; (ii) dual-level hypergraph contrastive strategy; (iii) adaptive temperature enhanced contrastive optimization.

### 4.1 NOISE-ENHANCED HYPERGRAPH AUGMENTATION

Inspired by the existing works (You et al., 2020; Wei et al., 2022; Lee & Shin, 2023), we first summarize five types of hypergraph augmentation methods (i.e., hyperedge removal, edge perturbation, attribute masking, node dropping, and subgrpah) listed in Appendix Table 3. Given the hypergraph augmentation set $\mathcal{T}$ listed in the table, we randomly select one pair of hypergraph augmentation methods from $\mathcal{T}$ and further obtain the augmented hypergraph pair $(\mathcal{G}_1, \mathcal{G}_2)$. Inspired by the conclusion that relatively challenging contrastive learning tasks can enhance the ability of representation learning compared with easy contrastive learning tasks (You et al., 2020; Jiang et al., 2020; Qian et al., 2022), we propose to generate challenging augmented hypergraph pairs by performing random noise over the augmented hypergraphs. Specifically, with the augmented graph pair $(\mathcal{G}_1, \mathcal{G}_2)$, for each node $v_i \in \mathcal{V}$, we perform a random noise $\delta_i$ following a specific distribution (e.g., uniform distribution) to the node attribute feature $x_i$. The attribute feature with noise denoted as $\widetilde{\mathbf{X}}$ is formulated as $\widetilde{\mathbf{X}} = \mathbf{X} + \delta = [x_1 + \delta_1; x_2 + \delta_2; \cdots; x_N + \delta_N]$, where $\mathbf{X}$ is the original node attribute feature, $[\cdot; \cdot]$ is the concatenation operator among node attribute features, and $+$ is the element-wise operator to add the original attribute vector and the random noise vector. Afterward, we obtain two noise-enhanced hypergraphs $[\widetilde{\mathcal{G}}_1 = (\mathcal{V}_1, \mathcal{E}_1, \widetilde{\mathcal{X}}_1), \widetilde{\mathcal{G}}_2 = (\mathcal{V}_2, \mathcal{E}_2, \widetilde{\mathcal{X}}_2)]$, where $\mathcal{V}_*$ and $\mathcal{E}_*$ are the node sets and the hyperedge sets of the corresponding augmented hypergraph $\widetilde{\mathcal{G}}_*$.

### 4.2 DUAL-LEVEL HYPERGRAPH CONTRASTIVE STRATEGY

After obtaining the noise-enhanced augmented hypergraphs, a dual-level hypergraph contrastive strategy is devised to align the node embeddings in a local manner and match the group-wise community embeddings from a global perspective, such that we can gain powerful representations (i.e., node embeddings and hyperedge embeddings) over the unlabeled data.

#### 4.2.1 NODE-LEVEL HYPERGRAPH CONTRASTIVE LEARNING

Following existing works (e.g., HyperGCL (Wei et al., 2022)) that aim to achieve the node embedding agreements by maximizing the similarity between positive node pairs in hypergraphs while minimizing the similarity between negative node pairs, we also employ a node-level hypergraph contrastive learning (HyGCL) module to ensure that the same nodes from different augmented hypergraphs are encoded closely, while different nodes are embedded farther apart. Specifically, given two nodes $(v_i, v_j)$ from $(\widetilde{\mathcal{G}}_1, \widetilde{\mathcal{G}}_2)$, we obtain the node embeddings $(\mathbf{u}_i^1, \mathbf{u}_j^2)$ by feeding the augmented graph $(\widetilde{\mathcal{G}}_1, \widetilde{\mathcal{G}}_2)$ to the encoder $f_{\mathcal{V} \rightarrow \mathcal{E} \rightarrow \mathcal{V}}$ in Equation 1. We then feed $(\mathbf{u}_i^1, \mathbf{u}_j^2)$ to projection head $h(\cdot)$. $(v_i, v_j)$ is a positive contrastive pair if $i = j$. Otherwise, it is a negative pair in node-level HyGCL.

#### 4.2.2 COMMUNITY-LEVEL HYPERGRAPH CONTRASTIVE LEARNING

Although the node-level HyGCL captures the information of individual nodes, it may not be sufficient for capturing the collective node behaviors within hyperedges. To this end, TriCL (Lee & Shin, 2023) proposes a group-level contrast strategy that aims to achieve the agreements among hyperedge embeddings $\mathbf{Z}$ for capturing the hyperedge behaviors. However, hyperedge embeddings $\mathbf{Z}$ still have limitations in capturing the collective node behaviors within hyperedges. Detailed explanations are provided in Proposition 1. In light of this, we design a community-level HyGCL module to capture the group behaviors within hyperedges from a global perspective. Specifically, we first introduce "community" to describe the collective node behaviors within hyperedges comprehensively. For each hyperedge $e_i \in \mathcal{E}$, we first get the hyperedge embedding $\mathbf{z}_i$ by applying the hypergraph encoder $f_{\mathcal{V} \rightarrow \mathcal{E}}$ in Equation 1. Then the community embeddings $\mathbf{H}$ integrating the edge embeddings $\mathbf{Z}$ and the node embeddings $\mathbf{U}$ within the corresponding hyperedges is formulated as:

$$\mathbf{h}_i = \mathbf{z}_i \oplus \frac{1}{d(e_i)} \sum_{m \in e_i} \mathbf{u}_m, \tag{3}$$

where $\mathbf{h}_i$ denotes the community embedding distinguied by hyperedge $e_i$, $\mathbf{z}_i$ denotes the corresponding hyperedge embedding, $\mathbf{u}_m$ represents the embedding of node $v_m$ within the hyperedge $e_i$, $d(e_i)$ denotes the degree of $e_i$, and $\oplus$ is the concatenation operator. Afterward, we leverage $\mathbf{h}_i^1$ and $\mathbf{h}_j^2$ to denote the community embedding distinguished by hyperedge $e_i$ and $e_j$ in the augmented graph $\widetilde{\mathcal{G}}_1$ and $\widetilde{\mathcal{G}}_2$, respectively. We further design a community-level contrastive strategy to reach agreements among community embeddings. In specific, similar to the node-level HyGCL, after feeding $(\mathbf{h}_i^1, \mathbf{h}_j^2)$ to a projection head $h(\cdot)$, we consider the community embeddings distinguished by the corresponding hyperedge from different augmented hypergraphs as the positive contrastive community pairs, and we expect that their community embeddings would stay closer than others. On the contrary, the community embeddings distinguished by different hyperedges from different augmented hypergraphs should be far apart. To make it clear, given a community embedding pair $(\mathbf{h}_i^1, \mathbf{h}_j^2)$ from $(\widetilde{\mathcal{G}}_1, \widetilde{\mathcal{G}}_2)$, $(\mathbf{h}_i^1, \mathbf{h}_j^2)$ will be viewed as positive contrastive pair if $i = j$. Otherwise, it would be a negative contrastive community pair. Next, we propose to prove that our community-level HyGCL excels in capturing group-wise behaviors.

**Proposition 1.** *Compared with hyperedge embeddings $\mathbf{Z}$, community embeddings $\mathbf{H}$ excel in capturing collective behaviors in hypergraphs.*

*Proof Sketch.* In Equation 3, $\frac{1}{d(e_i)} \sum_{m \in e_i} \mathbf{u}_m$ captures the collective behaviors among nodes by averaging their embeddings and $\mathbf{z}_i$ ensures that the community embedding incorporates the hyperedge's characteristics. Let us show an example: Consider two hyperedges $(e_i^1, e_j^2)$ from augmented graphs $(\widetilde{\mathcal{G}}_1, \widetilde{\mathcal{G}}_2)$ where $e_i^1$ contains nodes $\{v_1, v_2, v_3, v_4\}$ and $e_j^2$ contains nodes $\{v_3, v_4, v_7\}$, as illustrated in community-level hypergraph contrastive learning of Figure 1. Contrastive learning on hyperedge embeddings might be challenging to distinguish negative hyperedge pair $\mathbf{z}_i^1$ and $\mathbf{z}_j^2$, as they share most of the nodes during information propagation. In contrast, the community embeddings $\mathbf{h}_i^1$ and $\mathbf{h}_j^2$ provide more distinguishable representations that integrate both hyperedge characteristics and the node embeddings within hyperedges, making them more informative and effective for distinguishing negative community pairs. Therefore, community embeddings $\mathbf{H}$ excel in capturing collective behaviors in hypergraph compared to hyperedge embeddings $\mathbf{Z}$. □

### 4.3 Adaptive Temperature Enhanced Hypergraph Contrastive Learning

To make positive contrastive pairs closer and negative contrastive pairs farther, several contrastive losses are designed, such as NT-Xent loss (Chen et al., 2020) and JSD loss (Wang & Isola, 2020). All contrastive losses occupy the temperature index $\tau$ as the proxy to scale the embeddings and further control the penalties on negative samples. However, most consider temperature as a hyper-parameter to scale the representations but ignore the fact that a fixed temperature may not be optimal during the whole training process. Next, we first prove the importance of temperature in contrastive learning and further design an adaptive temperature learning module to enhance our dual-level HyGCL.

#### 4.3.1 Temperature Analysis in Contrastive Learning

Next, we would like to discuss and understand the influence of temperature in contrast optimization.

**Proposition 2.** *Contrastive loss is a hardness-aware loss.*

*Proof Sketch.* We first analyze the gradients of the contrastive loss w.r.t. positive and negative contrastive pairs. Here we take NT-Xent loss in Equation 2 as an example. According to the equation, the gradients w.r.t. the similarity of positive pair $s_{i,i}$ and negative pair $s_{i,j}$ are formulated as:

$$
\begin{aligned}
\frac{d\mathcal{L}_{NT}}{dS_{i,i}} &= -\frac{1}{\tau} \sum_{k \neq i} \frac{\exp\left(S_{i,k}/\tau\right)}{\sum_{l \neq k} \exp\left(S_{k,l}/\tau\right) + \exp\left(S_{i,k}/\tau\right)}, \\
\frac{d\mathcal{L}_{NT}}{dS_{i,j}} &= \frac{1}{\tau} \frac{\exp\left(S_{i,j}/\tau\right)}{\sum_{i \neq k} \exp\left(S_{i,k}/\tau\right) + \exp\left(S_{i,j}/\tau\right)},
\end{aligned}
\tag{4}
$$

where $S_{i,j}$ is the embedding similarity of contrastive pairs, and $j \neq i$. From the above equation, on the one hand, we find out that the temperature index $\tau$ controls the distribution of the gradients and the gradient of negative contrastive pairs is proportional to $S_{i,j}/\tau$, which demonstrates that the contrastive loss is hardness-aware as harder negative samples (i.e., having higher similarities) have larger gradients. On the other hand, the gradient of positive contrastive pairs will be affected by the

negative contrastive pairs and the magnitude of gradient w.r.t. the positive contrastive pairs equals the sum of gradients w.r.t. all negative contrastive pairs (i.e., $|\frac{d\mathcal{L}_{NT}}{dS_{i,i}}| = \sum_{k \neq i} |\frac{d\mathcal{L}_{NT}}{dS_{i,k}}|$). ☐

**Proposition 3.** *Temperature index $\tau$ controls the penalties on hard negative contrastive pairs.*

*Proof Sketch.* Please refer to Appendix B for the detailed proof. ☐

### 4.3.2 ADAPTIVE TEMPERATURE IN DUAL-LEVEL HYPERGRAPH CONTRASTIVE LEARNING

The aforementioned justifications prove the importance of temperature in contrastive learning. However, most works consider the temperature as a hyper-parameter and tend to assign a smaller value (e.g., $\tau = 0.05$) in most scenarios, but ignoring the fact that a fixed value of $\tau$ may not be optimal during the training process. In light of this, we design a module where the temperature can be learned at an adjustable pace based on the distance among these negative contrastive pairs. If the distance among negative contrastive pairs is small (hard negative pairs), the temperature descents rapidly, while it descents slowly when the distance among negative pairs is large (easy negative pairs). Eventually, the temperature will converge to an optimal value. Formally, the adaptive module in dual-level HyGCL is formulated as:

$$\tau_{nd}^{(t)} = max\,\{\tau_{nd}^{(t-1)} - \eta[\,1/\log\frac{\sum\limits_{i}^{|\mathcal{V}|}\sum\limits_{j \neq i}^{|\mathcal{V}|} \exp\,(\,\rho * ||\mathbf{u}_i^{(t)_1} - \mathbf{u}_j^{(t)_2}||^2)}{|\mathcal{V}| * (|\mathcal{V}| - 1)}\,]\,, \tau_{low}\},$$

$$\tau_{cm}^{(t)} = max\,\{\tau_{cm}^{(t-1)} - \eta[\,1/\log\frac{\sum\limits_{i}^{|\mathcal{E}|}\sum\limits_{j \neq i}^{|\mathcal{E}|} \exp\,(\,\rho * ||\mathbf{h}_i^{(t)_1} - \mathbf{h}_j^{(t)_2}||^2)}{|\mathcal{E}| * (|\mathcal{E}| - 1)}\,]\,, \tau_{low}\}. \tag{5}$$

Here $\tau_{nd}^{(t)}$ is the adaptive temperature in node-level HyGCL at the current training epoch $t$. $\tau_{cm}^{(t)}$ is the adaptive temperature in community-level HyGCL. $||\mathbf{u}_i^{(t)_1} - \mathbf{u}_j^{(t)_2}||$ and $||\mathbf{h}_i^{(t)_1} - \mathbf{h}_j^{(t)_2}||$ are the pairwise distances among nodes and communities in the embedding space from different augmented hypergraphs $\widetilde{\mathcal{G}}_1$ and $\widetilde{\mathcal{G}}_2$ at time $t$. $\eta$ denotes the learning rate; $\rho$ is the scaling factor to control the influence of distance among embeddings; $\tau_{low}$ is the hyperparameter to control the lower bound of the temperature index. Mention that, the lower bound $\tau_{low}$ is to prevent the temperature from becoming too small or approaching zero, ensuring a more reliable learning process. More discussions about $\tau_{low}$ are discussed in experiments. $\tau_{nd}^{(0)}$ and $\tau_{cm}^{(0)}$ are set as 1.0 in this work.

**Proposition 4.** *Temperature index $\tau_{nd}^{(t)}$ and $\tau_{cm}^{(t)}$ are adaptive to negative contrastive pairs during hypergraph contrastive optimization.*

*Proof Sketch.* Equation 5 is designed to adaptively adjust the temperature in hypergraph contrastive optimization. We leverage $\phi^{(t)}$ to learn the relative distance among negative contrastive pairs:

$$\phi_{nd}^{(t)} = 1/\log\frac{\sum\limits_{i}^{|\mathcal{V}|}\sum\limits_{j \neq i}^{|\mathcal{V}|} \exp\,(\,\rho * ||\mathbf{u}_i^{(t)_1} - \mathbf{u}_j^{(t)_2}||^2)}{|\mathcal{V}| * (|\mathcal{V}| - 1)}. \tag{6}$$

Here, we take node-level HyGCL as an example. The distance $||\mathbf{u}_i^{(t)_1} - \mathbf{u}_j^{(t)_2}||^2$ quantifies how dissimilar or hard it is to distinguish the embedding of node $v_i$ and $v_j$ at epoch $t$. A smaller distance indicates the negative contrastive pair is harder to distinguish. Then the exponential term amplifies the differences in distance. When the distance of negative pair is smaller, the exponential term $\exp(\rho * ||\mathbf{u}_i^{(t)_1} - \mathbf{u}_j^{(t)_2}||^2)$ will be closer to 1, resulting in a smaller value inside the logarithm and further a larger value of the inverse logarithm $\phi_{nd}^{(t)}$. Otherwise, the logarithm increases and $\phi_{nd}^{(t)}$ decreases, when the distance is larger. Therefore, according to Equation 5, when the distances among negative contrastive node pairs are small (indicating hard negative pairs), $\tau_{nd}^{(t)}$ will descent rapidly because it subtracts a large $\phi_{nd}^{(t)}$ from $\tau_{nd}^{(t-1)}$. Conversely, when the distances among negative node pairs are large (indicating easy negative pairs), $\tau_{nd}^{(t)}$ will descent more slowly because it subtracts a smaller $\phi_{nd}^{(t)}$ from $\tau_{nd}^{(t-1)}$. ☐

### 4.3.3 OVERALL OPTIMIZATION

With the designed adaptive temperature in both node-level and community-level AdT-HyGCL, the overall contrastive loss $\mathcal{L}_{nd\_cm}$ can be formulated as:

$$\mathcal{L}_{nd\_cm} = \lambda_1 * \mathcal{L}_{nd} + \lambda_2 * \mathcal{L}_{cm}, \text{ where}$$

$$\mathcal{L}_{nd} = -\log \sum_{v_i \in \mathcal{V}} \frac{\exp\left(\text{sim}\left(\mathbf{u}_i^1, \mathbf{u}_i^2\right)/\tau_{nd}^*\right)}{\sum_{k \neq i} \exp\left(\text{sim}\left(\mathbf{u}_i^1, \mathbf{u}_k^2\right)/\tau_{nd}^*\right) + \exp\left(\text{sim}\left(\mathbf{u}_i^1, \mathbf{u}_i^2\right)/\tau_{nd}^*\right)},$$

$$\mathcal{L}_{cm} = -\log \sum_{e_i \in \mathcal{E}} \frac{\exp\left(\text{sim}\left(\mathbf{h}_i^1, \mathbf{h}_i^2\right)/\tau_{cm}^*\right)}{\sum_{k \neq i} \exp\left(\text{sim}\left(\mathbf{h}_i^1, \mathbf{h}_k^2\right)/\tau_{cm}^*\right) + \exp\left(\text{sim}\left(\mathbf{h}_i^1, \mathbf{h}_i^2\right)/\tau_{cm}^*\right)}.$$

(7)

Here $\lambda_1$ and $\lambda_2$ are hyper-parameters to balance node-level and community-level AdT-HyGCL. $\tau_{nd}^*$ and $\tau_{cm}^*$ are the adaptive temperatures at both levels. To validate the generalization of temperature-enhanced contrast optimization, we also apply it to JSD (Jensen-Shannon divergence) loss for comparison in Table 1. The pseudo-code of AdT-HyGCL is listed in Appendix Algorithm 1.

## 5 EXPERIMENTS

In this section, we first introduce the experimental setup including datasets, baseline methods, and experimental settings. We then compare AdT-HyGCL with various baseline methods to show its effectiveness and robustness. Moreover, systemic studies about hypergraph augmentations and temperature index are conducted to show the rationality, generalization, and effectiveness of AdT-HyGCL. More details about data statistics (Appendix C), baseline settings (Appendix D), ablation study (Appendix E), performances of AdT-HyGCL with various augmentations (Appendix F), and complexity analysis (Appendix G) are also provided in Appendix.

### 5.1 EXPERIMENTAL SETUP

**Dataset.** We employ eight benchmark datasets from existing HyGNNs literature, including three co-citation and co-authorship networks from (Yadati et al., 2019) (i.e., Cora, Citerseer, Cora-CA), two hypergraph datasets (i.e., Zoo and Mushroom) from the UCI categorical machine learning repository (Asuncion & Newman, 2007), one computer vision hypergraph data (i.e., NTU2012) from (Chen et al., 2003), and two e-commerce hypergraph networks from (Chien et al., 2022) (i.e., House and Walmart). More detailed discussion and data statistics are introduced in Appendix C.

**Baseline Methods.** To evaluate the performance of AdT-HyGCL, we consider nine baseline methods including six HyGNNs (i.e., CEGCN (Feng et al., 2019), HNHN (Dong et al., 2020), HGNN (Feng et al., 2019), HCHA (Bai et al., 2021), UniGCNII (Huang & Yang, 2021), and AllDeepSets (Chien et al., 2022)) and three recent contrastive learning methods over hypergraphs, i.e., HyperGCL (Wei et al., 2022), CHGNN (Song et al., 2023), and TriCL (Lee & Shin, 2023). Details about baseline methods are introduced in Appendix D.

**Experimental Settings.** All experiments are conducted under the environment of the Ubuntu 16.04 OS, plus Intel i9-9900k CPU, two GeForce GTX 2080 Ti Graphics Cards, and 64 GB of RAM. To make fair comparisons, we exactly follow the settings of HyperGCL: (i) We train all methods with 500 epochs; (ii) The train/val/test ratio is 10%/10%/80%; (iii) We adopt AllDeepSets (Chien et al., 2022) as the encoder over all datasets. Besides, we adopt two metrics, i.e., accuracy and Macro-F1 to evaluate all models. Moreover, all models are trained five times, and the average performance multiplied by 100 on testing data is reported. In Equation 5, $\tau_{low}$ is set as 0.05 over all datasets. $\eta$ and $\rho$ are set as 0.001 and 0.5, respectively. The trade-off hyper-parameters $\lambda_1$ and $\lambda_2$ are set as 1.0.

### 5.2 EXPERIMENT ANALYSIS

**AdT-HyGCL Enhances Semi-supervised Learning.** Table 1 shows the accuracy and Macro-F1 performance of all methods over eight datasets for node classification tasks. From this table, we conclude that: (i) contrastive learning over unlabeled data boosts the representation learning in hypergraphs as all contrastive learning methods outperform the corresponding hypergraph encoder. (ii) our proposed model AdT-HyGCL outperforms most SOTAs of HyGNNs and hypergraph contrastive learning, showing the effectiveness of AdT-HyGCL in enhancing hypergraph representation learning; (iii) AdT-HyGCL with different contrastive losses (i.e., JSD and NT-Xent loss) gains excellent performance, showing its excellent generalization with different contrast loss optimizations.

**AdT-HyGCL Boosts Model Robustness.** We further conduct experiments to demonstrate that AdT-HyGCL boosts the model robustness. Specifically, we perform two types of attacks including min-

Table 1: Performance comparison (Mean % ± std) of all methods for node classification. Purple shaded numbers indicate the best result and gray shade numbers represent the runner-up performance. If the best result comes from AdT-HyGC, we shade the best baseline result in gray.

| | Cora | Citeseer | Cora-CA | Zoo | Mushroom | NTU2012 | House | Walmart |
|---|---|---|---|---|---|---|---|---|
| *Accuracy* | | | | | | | | |
| CEGCN | 68.11 ± 1.67 | 62.20 ± 0.60 | 68.69 ± 1.91 | 63.34 ± 2.53 | 94.30 ± 0.24 | 65.70 ± 3.46 | 56.69 ± 2.56 | 49.23 ± 0.23 |
| HNHN | 64.96 ± 2.41 | 62.38 ± 1.52 | 68.01 ± 5.61 | 99.64 ± 0.26 | 65.06 ± 1.85 | 59.02 ± 1.61 | 42.24 ± 0.37 |
| HGNN | 66.98 ± 0.89 | 56.63 ± 1.96 | 73.71 ± 1.50 | 69.21 ± 11.39 | 98.17 ± 0.31 | 69.05 ± 1.50 | 54.30 ± 1.46 | 55.15 ± 0.28 |
| HCHA | 72.20 ± 1.69 | 65.33 ± 0.57 | 75.12 ± 0.99 | 69.51 ± 10.02 | 97.65 ± 0.45 | 73.88 ± 1.74 | 55.73 ± 1.68 | 60.16 ± 0.22 |
| UniGCNII | 71.23 ± 0.66 | 65.71 ± 1.48 | 77.35 ± 0.27 | 69.09 ± 10.64 | 99.85 ± 0.04 | 74.27 ± 1.41 | 63.27 ± 1.48 | 49.48 ± 0.41 |
| AllDeepSets | 68.14 ± 1.31 | 63.60 ± 1.27 | 68.52 ± 2.67 | 58.48 ± 9.13 | 99.72 ± 0.18 | 72.54 ± 1.42 | 58.74 ± 2.93 | 55.89 ± 0.24 |
| HyperGCL | 73.52 ± 1.05 | 66.82 ± 0.98 | 76.57 ± 1.70 | 58.77 ± 6.09 | 99.76 ± 0.15 | 76.16 ± 1.26 | 58.30 ± 4.18 | 60.33 ± 0.22 |
| CHGNN | 74.20 ± 0.58 | 68.83 ± 1.83 | 77.09± 1.01 | 65.12 ± 3.39 | 94.64 ± 0.72 | 74.34 ± 1.18 | 59.26 ± 1.52 | — |
| TriCL | 64.86 ± 0.80 | 62.88 ± 1.51 | 77.21 ± 1.42 | 64.25 ± 9.23 | 95.53 ± 1.31 | 75.01 ± 1.76 | 58.46 ± 1.78 | 59.48 ± 0.50 |
| AdT-HyGCL (JSD) | 73.75 ± 1.21 | 67.14 ± 1.27 | 79.35 ± 1.70 | 69.88 ± 12.45 | 99.82 ± 0.13 | 77.54 ± 0.77 | 59.10 ± 1.80 | 60.55 ± 0.28 |
| AdT-HyGCL (NT) | 76.08 ± 0.99 | 69.74 ± 0.62 | 79.24 ± 1.69 | 69.14 ± 13.12 | 99.92 ± 0.05 | 77.64 ± 0.84 | 60.62 ± 1.07 | 60.82 ± 0.34 |
| *Macro-F1* | | | | | | | | |
| CEGCN | 64.68 ± 2.92 | 56.13 ± 0.78 | 66.27 ± 2.36 | 14.28 ± 3.80 | 94.28 ± 0.24 | 52.09 ± 4.71 | 52.83 ± 4.70 | 20.44 ± 0.71 |
| HNHN | 61.12 ± 2.28 | 56.24 ± 1.91 | 62.15 ± 1.20 | 41.85 ± 16.08 | 99.64 ± 0.26 | 50.70 ± 2.49 | 58.97 ± 1.61 | 13.24 ± 0.78 |
| HGNN | 63.91 ± 0.87 | 56.13 ± 0.78 | 71.19 ± 2.34 | 43.14 ± 12.19 | 98.16 ± 0.31 | 55.62 ± 2.41 | 52.65 ± 2.90 | 25.43 ± 0.23 |
| HCHA | 68.89 ± 2.79 | 56.24 ± 1.91 | 71.45 ± 2.80 | 40.37 ± 13.89 | 97.65 ± 0.45 | 60.98 ± 4.18 | 54.99 ± 2.20 | 42.62 ± 0.15 |
| UniGCNII | 68.24 ± 1.31 | 58.43 ± 1.90 | 74.64 ± 0.79 | 38.20 ± 9.99 | 99.85 ± 0.04 | 60.90 ± 2.64 | 63.06 ± 1.49 | 23.24 ± 0.39 |
| AllDeepSets | 64.84 ± 1.18 | 59.36 ± 0.83 | 59.36 ± 0.83 | 24.66 ± 10.27 | 99.72 ± 0.18 | 59.19 ± 2.85 | 58.12 ± 2.60 | 26.32 ± 0.27 |
| HyperGCL | 71.27 ± 0.84 | 62.69 ± 1.40 | 74.28 ± 1.85 | 26.09 ± 6.09 | 99.76 ± 0.15 | 64.66 ± 2.48 | 57.70 ± 4.43 | 41.69 ± 0.86 |
| CHGNN | 72.34 ± 0.72 | 64.98 ± 1.77 | 75.27 ± 1.20 | 37.94 ± 2.38 | 94.66 ± 0.86 | 60.47 ± 1.36 | 56.96 ± 0.79 | — |
| TriCL | 72.15 ± 1.45 | 63.41 ± 1.23 | 74.34 ± 1.52 | 30.45 ± 10.24 | 96.43 ± 1.71 | 62.42 ± 1.51 | 56.90 ± 1.74 | 40.39 ± 0.71 |
| AdT-HyGCL (JSD) | 74.10 ± 1.58 | 64.65 ± 0.85 | 77.40 ± 2.32 | 42.59 ± 8.14 | 99.83 ± 0.32 | 66.68 ± 1.44 | 59.72 ± 1.37 | 43.58 ± 0.75 |
| AdT-HyGCL (NT) | 74.47 ± 0.89 | 64.48 ± 1.25 | 77.42 ± 1.77 | 40.57 ± 17.57 | 99.93 ± 0.05 | 66.47 ± 1.89 | 60.22 ± 0.90 | 42.23 ± 0.83 |

Table 2: Performance comparison in terms of model robustness. The best performance is shaded in purple and the runner-up is shaded in gray.

| Setting | Cora | | | | Citseer | | | |
|---|---|---|---|---|---|---|---|---|
| | Minmax | | Nettack | | Minmax | | Nettack | |
| Model | Accuracy | Macro-F1 | Accuracy | Macro-F1 | Accuracy | Macro-F1 | Accuracy | Macro-F1 |
| UniGCNII | 70.04 ± 1.18 | 67.21 ± 1.87 | 69.45 ± 1.24 | 66.57 ± 1.38 | 62.17 ± 4.38 | 60.15 ± 1.32 | 65.38 ± 1.57 | 59.02 ± 1.68 |
| AllDeepSets | 66.94 ± 1.11 | 63.81 ± 1.17 | 67.01 ± 1.25 | 62.25 ± 1.03 | 62.56 ± 1.05 | 55.97 ± 1.32 | 64.30 ± 1.17 | 59.98 ± 1.58 |
| HyperGCL | 69.99 ± 0.71 | 67.44 ± 0.86 | 70.77 ± 0.84 | 68.39 ± 0.69 | 64.99 ± 1.89 | 61.00 ± 3.10 | 65.25 ± 1.02 | 61.46 ± 1.09 |
| AdT-HyGCL | 75.11 ± 0.52 | 73.01 ± 0.82 | 75.03 ± 1.36 | 73.24 ± 1.44 | 68.84 ± 0.91 | 63.53 ± 0.48 | 69.30 ± 0.93 | 64.36 ± 1.01 |

| Setting | NTU2012 | | | | House | | | |
|---|---|---|---|---|---|---|---|---|
| | Minmax | | Nettack | | Minmax | | Nettack | |
| Model | Accuracy | Macro-F1 | Accuracy | Macro-F1 | Accuracy | Macro-F1 | Accuracy | Macro-F1 |
| UniGCNII | 68.76 ± 2.05 | 58.23 ± 1.70 | 72.95 ± 2.23 | 62.17 ± 4.38 | 61.05 ± 1.84 | 59.41 ± 1.63 | 60.60 ± 1.29 | 60.02 ± 1.49 |
| AllDeepSets | 67.17 ± 2.60 | 53.05 ± 4.22 | 71.83 ± 1.94 | 57.57 ± 2.06 | 55.38 ± 2.32 | 54.12 ± 2.92 | 58.86 ± 3.03 | 57.13 ± 4.53 |
| HyperGCL | 71.88 ± 0.46 | 60.43 ± 2.08 | 74.21 ± 1.43 | 62.54 ± 1.71 | 57.62 ± 2.00 | 56.39 ± 3.17 | 59.65 ± 1.62 | 59.02 ± 2.11 |
| AdT-HyGCL | 71.95 ± 0.65 | 61.42 ± 2.24 | 74.75 ± 2.24 | 62.99 ± 2.63 | 59.78 ± 2.22 | 59.95 ± 3.20 | 59.19 ± 1.05 | 60.11 ± 0.81 |

max attack (Sun et al., 2020) and nettack (Zügner et al., 2018) over four real-world hypergraphs. From Table 2, we observe that both minmax and nettack attacks affect the performance, while our mode AdT-HyGCL is more robust compared with other baseline models. The performance of AdT-HyGCL shows a relatively small decline compared with results in Table 1, showing its robustness.

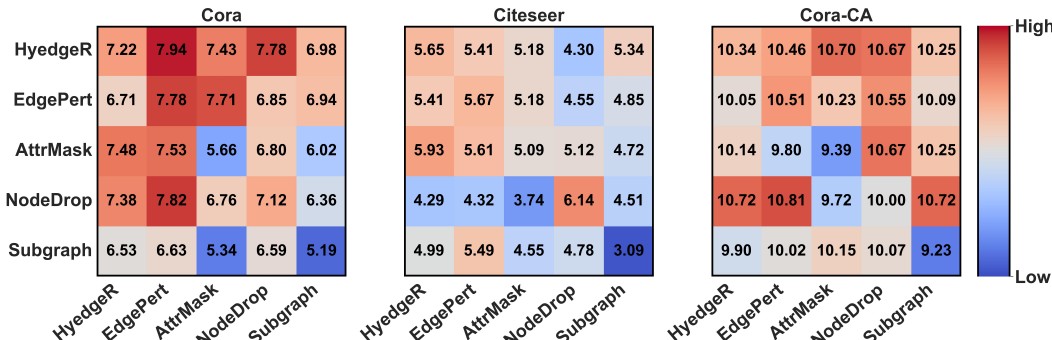

Figure 2: Accuracy gain (%) of AdT-HyGCL with different augmentations over node classification, compared with AllDeepSets. Warmer colors show more performance gains while cooler colors represent less performance gains.

**Unveil the Effectiveness of Hyperedge Removing Augmentation.** Hyperedge removing (HyedgeR) augmentation is explicitly designed for hypergraphs. Please refer to Appendix Table 3 for augmentation abbreviations. Figure 2 illustrates the results of AdT-HyGCL with various augmentations over three datasets. Table 6 in the Appendix shows the results of AdT-HyGCL with various augmentation combinations over all datasets. As the first row of each subfigure in Figure 2 shows, augmentation incorporates HyedgeR benefit hypergraph representation learning as most of them gain better performance with warm colors. Besides, from Appendix Table 6, we find out most of the best performances are involved with HyedgeR, which is shaded in purple. Based on the above findings, we demonstrate that randomly removing partial hyperedges would not alter global semantics and can enhance hypergraph contrastive learning.

**Synergistic Effects of Augmentation Combination.** We also conduct extensive experiments to study the synergistic effects among hypergraph augmentations in the semi-supervised setting. As Figure 2 illustrated, we conclude that the best performances mostly come from the combinations of different types of augmentations. Based on this, we further visualize contrastive loss curves composing augmentations together with HyedgeR over three datasets in Figure 3. From this figure, we conclude that contrastive loss with the same type of augmentations (HyedgeR+HyperedgeR in blue) always descents faster than that with different types of augmentations (e.g., HyedgeR+NodeDrop in red), showing that different types of augmentations might benefit more in hypergraph learning.

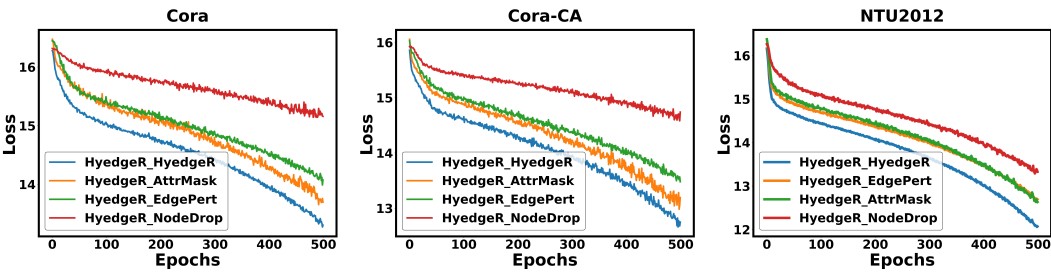

Figure 3: Contrastive loss curves for different augmentation pairs involved with HyedegeR.

**Temperature Analysis in AdT-HyGCL.** Besides theoretical justifications, we conducted additional experiments to study the influence of temperature on AdT-HyGCL. Figure 4 illustrates the performance of AdT-HyGCL with static $\tau$, and AdT-HyGCL without the lower bound constraint $\tau_{low}$. We find out that the static values of $\tau$ achieve excellent performance but still do not yield the best performances. Additionally, the performance of AdT-HyGCL without $\tau_{low}$ decreases obviously over these datasets. These findings show the rationality of our adaptive contrast optimization.

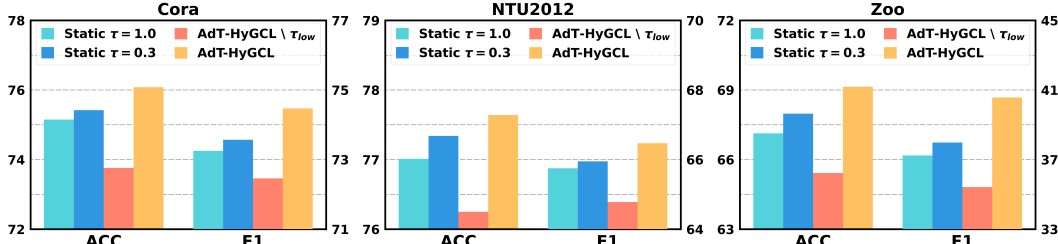

Figure 4: Performance of AdT-HyGCL under different contrast optimization strategies over Cora, NTU2012, and Zoo. The left axis is for accuracy and the right axis is for Macro-F1.

## 6 CONCLUSION

The paper introduces an adaptive temperature-enhanced dual-level hypergraph contrastive learning model called AdT-HyGCL to enhance hypergraph contrastive learning. To handle the limitations of existing hypergraph contrastive learning w.r.t. underestimating group-wise behaviors and ignoring the importance of temperature in contrast optimization, AdT-HyGCL introduces a dual-level contrast mechanism to capture individual behaviors and group-wise behaviors simultaneously. Besides, it designs adaptive temperature-enhanced contrast optimization to improve discrimination ability between contrastive pairs, thus boosting hypergraph representation learning over unlabeled data. Theoretical justifications and empirical experiments on eight benchmark hypergraphs demonstrate the rationality, effectiveness, generalization, and robustness of AdT-HyGCL.

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

APPENDIX

## A  HYPERGRAPH AUGMENTATION

In this section, we would like to first summarize the hypergraph augmentation methods and then discuss the differences between hyperedge removal and edge perturbation.

First of all, we summarize five types of hypergraph augmentation methods in Table 3, i.e., hyperedge removing, edge perturbation, attribute masking, node dropping, and subgraph. The augmentation methods of attribute masking, node dropping, and subgraph in hypergraphs are similar to that in graph augmentation (Zhu et al., 2021a; You et al., 2020; Zhu et al., 2020) and we will focus more on discussing the differences between hyperedge removing and edge perturbation.

A hypergraph can be converted into an equivalent bipartite graph through a simple transformation process. In a hypergraph, the edges, known as hyperedges, can connect any number of nodes, creating a more generalized representation of relationships. To convert it into a bipartite graph, each hyperedge in the hypergraph becomes a node. As illustrated in Figure 5.(c), for every connection between a hyperedge and a node in the hypergraph, we create an edge in the bipartite graph that connects the corresponding nodes. This transformation captures the relationships between hyperedges and nodes in a bipartite graph.

Then, we would like to introduce the difference between hyperedge removal in the hypergraph and edge perturbation in the bipartite graph, which is illustrated in Figure 5. (a) and Figure 5.(d), respectively. From Figure 5.(a), hyperedge $e_2$ is removed from the hypergraph. We believe that removing partial hyperedges would not alter the full semantics. For instance, removing hyperedge $e_2$ will not alter the full semantics of nodes 3, 4, and 7 for sure. On the other hand, we perturb the connections in the bipartite graph including dropping edges and adding edges, as illustrated in Figure 5. (d). We believe that it can enhance the structure robustness and would not change the full semantic structure.

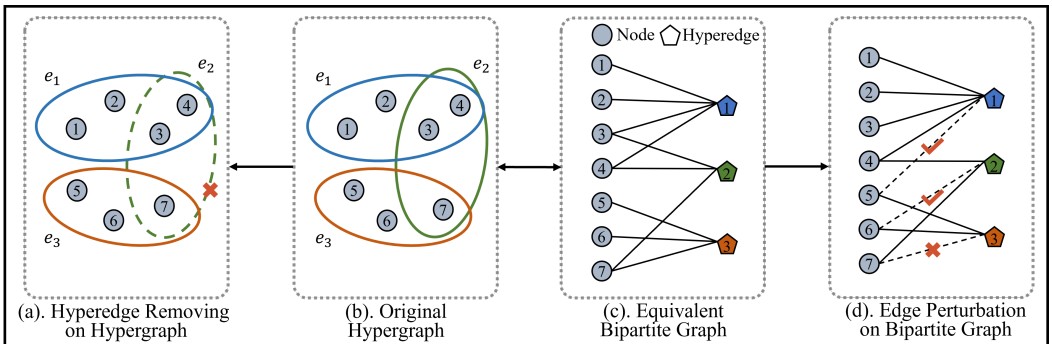

Figure 5: Subfigure (b) and (c) show the conversion between hypergraph and bipartite graph. Subgraph (a) illustrates how to remove hyperedges on hypergraph (HyedgeR); Subfigure (d) shows how to perturb (including add/remove) edges over the equivalent bipartite graph (EdgePert).

Table 3: Summary of hypergraph augmentation methods for hypergraph contrastive learning.

| Data augmentation | Abbreviation | Type | Underlying Prior |
|---|---|---|---|
| Hyperedge removing | HyedgeR | Hyperedges | Partial high-order relations missing does not alter full semantics. |
| Edge perturbation | EdgePert | Edges | Semantic robustness against the pair-wise connections noise. |
| Attribute masking | AttrMask | Nodes | Semantic robustness against partial node attributes missing. |
| Node dropping | NodeDrop | Nodes, Edges | Partial nodes missing does not change the global semantics. |
| Subgraph | Subgraph | Nodes, Hyperedges | Local structure can infer the full semantics. |

## B  TEMPERATURE ANALYSIS IN CONTRASTIVE LEARNING

As we proved in Proposition 2, contrastive loss is a hardness-aware loss that aims to align positive contrastive pairs while separating negative contrastive pairs. Next, we would like to prove that the temperature index controls the penalties on hard negative contrastive pairs.

**Proposition 3.** *Temperature index $\tau$ controls the penalties on hard negative contrastive pairs. A smaller temperature index puts larger penalties on hard negative contrastive pairs, while a larger temperature index tends to assign uniform penalties over negative contrastive pairs.*

*Proof Sketch.* As discussed, the gradients w.r.t. the similarity of positive pair $s_{i,i}$ and negative pair $s_{i,j}$ can be formulated as:

$$\begin{aligned}
\frac{\mathrm{d}\mathcal{L}_{NT}}{\mathrm{d}S_{i,i}} &= -\frac{1}{\tau} \sum_{k \neq i} \frac{\exp\left(S_{i,k}/\tau\right)}{\sum_{l \neq k} \exp\left(S_{k,l}/\tau\right) + \exp\left(S_{i,k}/\tau\right)}, \\
\frac{\mathrm{d}\mathcal{L}_{NT}}{\mathrm{d}S_{i,j}} &= \frac{1}{\tau} \frac{\exp\left(S_{i,j}/\tau\right)}{\sum_{i \neq k} \exp\left(S_{i,k}/\tau\right) + \exp\left(S_{i,j}/\tau\right)},
\end{aligned} \tag{8}$$

where $S_{i,j}$ is the embedding similarity of contrastive pairs, and $j \neq i$. From the above equation, we find out that the gradient of the negative contrastive pair $(v_i, v_j)$ is dominated by the temperature index $\tau$ and the embedding similarity $S_{i,j}$, as the denominator is fixed. With a smaller $\tau$, contrastive loss $\mathcal{L}_{NT}$ assigns larger gradients (larger penalties) to hard negative contrastive pairs as the numerator is larger. In this case, node $v_j$ with higher similarity would be farther away from the anchor node $v_i$ in a quick manner, while node $v_j$ with less similarity would get away from node $v_i$ slowly.

---

**Algorithm 1:** Training Procedure of AdT-HyGCL

**Data:** Hypergraph $\mathcal{G}$, Hypergraph augmentation set $\mathcal{T}$, Hypergraph encoder $f(\cdot)$, Projection head $h(\cdot)$.

**Result:** Pre-trained hypergraph encoder.

1 Randomly select two hypergraph augmentations $A_1$ and $A_2$ from $\mathcal{T}$.

2 **for** *each epoch $t$* **do**

3      Augmentation: $\mathcal{G} \xrightarrow{A_1} \widetilde{\mathcal{G}}_1, \mathcal{G} \xrightarrow{A_2} \widetilde{\mathcal{G}}_2$.

4      Random noise enhanced augmentation: $[\widetilde{\mathcal{G}}_1 = (\mathcal{V}_1, \mathcal{E}_1, \mathcal{X}_1 + \delta_1), \widetilde{\mathcal{G}}_2 = (\mathcal{V}_2, \mathcal{E}_2, \mathcal{X}_2 + \delta_2)]$.

5      Feed the augmented hypergraphs $(\widetilde{\mathcal{G}}_1, \widetilde{\mathcal{G}}_1)$ to $f(\cdot)$ for obtaining the node embeddings $(\mathbf{u}^1, \mathbf{u}^2)$ from the local view and the community embeddings $(\mathbf{h}^1, \mathbf{h}^2)$ from the global perspective ;

6      Feed the node and community embeddings into the projection head $h(\cdot)$;

7      Dynamically adjust the adaptive temperature index $\tau_{cm}^{(t)}$ at the node level and $\tau_{cm}^{(t)}$ at the community level via Eq. 5.

8      Optimize $f(\cdot)$ and $h(\cdot)$ by minimizing dual-level contrastive loss $\mathcal{L}_{nd\_cm}$ in Eq. 7.

---

## C    Data Description

We employ eight benchmark datasets from existing hypergraph neural networks literature including three co-citation and co-authorship networks from (Yadati et al., 2019) (i.e., Cora, Citerseer, Cora-CA), two hypergraph data (i.e., Zoo and Mushroom) from the UCI categorical machine learning repository (Asuncion & Newman, 2007), one computer vision hypergraph data NTU2012 (Chen et al., 2003), and two e-commerce hypergraph networks, i.e. House and Walmart from (Chien et al., 2022). Mention that, three co-citation and co-authorship networks are differently constructed in the hypergraph domain. For instance, all papers cited by a paper are connected by a hyperedge in Cora. In these co-citation hypergraphs, the node features are the bag-of-words representations of the corresponding documents. For the Zoo hypergraph, the node features are a mix of categorical and numerical features about animals. In the Mushroom hypergraph, node features represent categorical descriptions of 23 species of mushrooms. About the computer vision NTU2012 hypergraph, the features extracted are via Group-View Convolutions Neural Network (GVCNN) (Feng et al., 2018) and Multi-View Convolutional Neural Network (MVCNN) (Su et al., 2015). Besides, we construct the hypergraph based on the setting in (Feng et al., 2019). For the House dataset, each node represents a member of the US House of Representatives and these hyperedges are formed by grouping members of the same committee. The node labels show the political party of representatives. For the Walmart dataset, each node shows the product that is listed at Walmart and each hyperedge shows that a bunch of products are purchased together. As the original datasets, House and Walmart, do not have any

features, following the previous work (Chien et al., 2022), we impute the Gaussian random vectors to the one-hot encoding of the labels as the attribute features. The feature dimension for House and Walmart is 100 and the noise standard deviation is set as 1.0 in this work. More details about each dataset are listed in Table 4.

Table 4: Statistics of eight hypergraph datasets. $d_e$ represents the hyperedge degree and $d_v$ denotes the node degree in hypergraphs.

|  | Cora | Citeseer | Cora-CA | Zoo | Mushroom | NTU2012 | House | Walmart |
|---|---|---|---|---|---|---|---|---|
| $|\mathcal{V}|$ | 2,708 | 3,312 | 2,708 | 101 | 8,124 | 2,012 | 1,290 | 88,860 |
| $|\mathcal{E}|$ | 1,579 | 1,079 | 1,072 | 43 | 298 | 2,012 | 341 | 69,906 |
| # feature | 1,433 | 3,703 | 1,433 | 16 | 22 | 100 | 100 | 100 |
| # class | 7 | 6 | 7 | 7 | 2 | 67 | 2 | 11 |
| avg $d_e$ | 3.03 | 3.2 | 4.28 | 39.93 | 136.31 | 5 | 34.72 | 6.59 |
| max $d_e$ | 5 | 26 | 43 | 93 | 1,808 | 5 | 81 | 25 |
| min $d_e$ | 2 | 2 | 2 | 1 | 1 | 5 | 1 | 2 |
| med $d_e$ | 3 | 2 | 3 | 40 | 72 | 5 | 40 | 5 |
| avg $d_v$ | 1.77 | 1.04 | 1.69 | 17 | 5 | 5 | 9.18 | 5.18 |
| max $d_v$ | 145 | 88 | 23 | 17 | 5 | 19 | 44 | 5,733 |
| min $d_v$ | 0 | 0 | 0 | 17 | 5 | 1 | 0 | 0 |
| med $d_v$ | 1 | 0 | 2 | 17 | 5 | 5 | 7 | 2 |

## D    BASELINE METHOD

To validate the effectiveness of AdT-HyGCL in modeling hypergraph representations over unlabeled data, we compare AdT-HyGCL with two groups of baseline methods, G1 (hypergraph representation learning models) and G2 (hypergraph contrastive learning models). G1 contains six hypergraph representation learning methods including CEGCN (Feng et al., 2019), HNHN (Dong et al., 2020), HGNN (Feng et al., 2019), HCHA (Bai et al., 2021), UniGCNII (Huang & Yang, 2021), and AllDeepSets (Chien et al., 2022). All architectures are implemented using the Pytorch Geometric library (PyG) (Fey & Lenssen, 2019). Specifically, CEGCN and CEGAT are executed directly from PyG. Similar to the baseline setting in (Chien et al., 2022), we also adapt the implementation of HGNN, HCHA, and HNHN from PyG. Mention that in the original implementation of HGNN, propagation is performed via matrix multiplication which is far less memory and computationally efficient compared to the implementation. Besides, we exactly follow the setting of UniGCNII and AllDeepSets to reproduce the experimental results. G2 includes three recent hypergraph contrastive learning methods, HyperGCL (Wei et al., 2022), CHGNN (Song et al., 2023), and TriCL (Lee & Shin, 2023). To make a fair comparison with HyperGCL, CHGNN, and TriCL, we leverage the same encoder AllDeepSets as the hypergraph backbone to learn the hyperedge and node embeddings, and further integrate the contrastive self-supervised setting and the semi-supervised setting to do the downstream node classification tasks.

## E    ABLATION STUDY

To show the effectiveness of different components in AdT-HyGCL, we conduct a set of ablation experiments over four datasets (i.e., Cora, Citeseer, Cora-CA, and NTU2012) and further analyze the contribution of each component (i.e., random noise $\delta$ on the augmented graph (C1), the community-level hypergraph contrastive learning (C2), and the adaptive temperature during contrast optimization (C3)) by removing it separately. The performance is listed in Table 5. First, we remove the random noise $\delta$ from the augmented hypergraphs, which means we leverage the hypergraph after augmentations (i.e., HyedgeR and NodeDrop) to reach the agreement between node embeddings from the local view and the agreement between community embeddings from the global view. From Table 5, we find out the performance of removing C1 decreases slightly, showing the effectiveness of random noise in enhancing the hypergraph augmentation. Besides, we remove the community-level contrastive module (C2) and we observe that the performance declines obviously, which empirically shows the effectiveness of community-level hypergraph contrastive learning in modeling the hypergraph representations over unlabeled data. Moreover, the performance of removing C3 (adaptive temperature) also shows an obvious decline over four datasets, validating the effectiveness of adaptive temperature in AdT-HyGCL.

Table 5: Performance of model variants. The train/val/test ratio is 10%:10%:80%. The best performance is shaded in purple. Here AdT-HyGCL refers to AdT-HyGCL with NT loss. C1 refers to the random noise on the augmented graph; C2 refers to the community-level hypergraph contrastive learning; C3 represents the adaptive temperature enhancement during contrast optimization.

| Setting | Cora | | Citeseer | | Cora-CA | | NTU2012 | |
|---|---|---|---|---|---|---|---|---|
| Model | Accuracy | Macro-F1 | Accuracy | Macro-F1 | Accuracy | Macro-F1 | Accuracy | Macro-F1 |
| – C1 | 75.54 ± 1.01 | 73.07 ± 0.87 | 68.15 ± 0.74 | 63.14 ± 1.08 | 78.42 ± 1.54 | 76.35 ± 1.64 | 76.97 ± 0.97 | 65.45 ± 1.51 |
| – C2 | 74.25 ± 1.25 | 73.15 ± 1.68 | 67.41 ± 1.27 | 63.55 ± 1.17 | 77.45 ± 1.53 | 75.58 ± 1.47 | 76.99 ± 1.35 | 65.54 ± 1.02 |
| – C3 | 74.53 ± 1.71 | 73.49 ± 1.53 | 67.54 ± 0.74 | 63.17 ± 1.32 | 77.32 ± 1.42 | 75.24 ± 1.68 | 76.87 ± 0.77 | 65.34 ± 1.78 |
| AdT-HyGCL | 76.11 ± 0.98 | 74.45 ± 0.85 | 69.75 ± 0.71 | 64.49 ± 1.24 | 79.35 ± 1.71 | 77.42 ± 1.75 | 77.64 ± 0.87 | 66.48 ± 1.91 |

## F AdT-HyGCL with Different Augmentations

Inspired by GraphCL (You et al., 2020), we also conduct extensive experiments to study the synergistic effects among hypergraph augmentations in the semi-supervised setting. The performance of the optimal augmentation combinations is shaded with purple and the worst augmentation combinations are shaded with gray in Table 6. As this table listed, we find out that the best performances over different hypergraphs mostly come from the combinations of different types of augmentations. while the same type of augmentation is more likely to gain relatively less performance. For instance, AdT-HyGCL with Subgraph+Subgraph augmentations has the lowest performance over Cora, Citeseer, and Cora-CA, and it gains the worst performance over Mushroom if it takes AttrMask+AttrMask as the augmentation method.

Table 6: The accuracy performance (mean % ± std) comparison of our model AdT-HyGCL with different augmentations. Purple shaded values indicate the best result and gray shaded values represent the worst performance. A1: HyedgeR, A2: EdgePert, A3: AttrMask, A4: NodeDrop, A5: Subgraph.

| | Cora | Citeseer | Cora-CA | Zoo | Mushroom | NTU2012 | House | Walmart |
|---|---|---|---|---|---|---|---|---|
| A1:A1 | 75.36 ± 1.19 | 69.25 ± 0.83 | 78.86 ± 1.56 | 65.43 ± 13.38 | 99.92 ± 0.05 | 74.96 ± 1.34 | 59.72 ± 0.71 | 59.94 ± 0.32 |
| A1:A2 | 76.08 ± 0.99 | 69.01 ± 1.02 | 78.57 ± 2.03 | 65.68 ± 9.93 | 99.90 ± 0.09 | 75.14 ± 0.87 | 59.73 ± 0.84 | 59.87 ± 0.13 |
| A1:A3 | 75.57 ± 0.77 | 68.78 ± 0.99 | 79.22 ± 1.18 | 64.94 ± 10.28 | 99.79 ± 0.22 | 74.45 ± 1.80 | 59.43 ± 0.68 | 60.82 ± 0.35 |
| A1:A4 | 75.92 ± 1.27 | 67.90 ± 1.03 | 79.19 ± 1.77 | 68.15 ± 12.87 | 99.82 ± 0.14 | 77.64 ± 0.84 | 59.45 ± 0.77 | 59.82 ± 0.15 |
| A1:A5 | 75.12 ± 0.80 | 68.94 ± 0.86 | 78.77 ± 2.35 | 69.14 ± 13.12 | 99.81 ± 0.12 | 75.28 ± 1.32 | 59.34 ± 0.89 | 59.88 ± 0.25 |
| A2:A2 | 75.92 ± 1.44 | 69.27 ± 0.94 | 79.03 ± 1.55 | 67.16 ± 9.90 | 99.80 ± 0.19 | 75.32 ± 1.35 | 59.34 ± 0.97 | 59.02 ± 0.34 |
| A2:A3 | 75.85 ± 1.47 | 68.78 ± 0.93 | 78.75 ± 1.48 | 66.91 ± 12.51 | 99.80 ± 0.28 | 74.55 ± 1.76 | 59.42 ± 0.74 | 59.87 ± 0.28 |
| A2:A4 | 74.99 ± 1.38 | 68.15 ± 0.85 | 79.07 ± 1.71 | 67.16 ± 11.99 | 99.80 ± 0.13 | 77.18 ± 1.18 | 59.76 ± 0.68 | 59.72 ± 0.33 |
| A2:A5 | 75.08 ± 2.13 | 68.45 ± 0.94 | 78.61 ± 2.01 | 64.69 ± 12.52 | 99.86 ± 0.07 | 75.50 ± 1.79 | 59.87 ± 0.73 | 59.64 ± 0.37 |
| A3:A3 | 73.80 ± 2.82 | 68.69 ± 0.22 | 77.91 ± 1.86 | 65.93 ± 12.33 | 99.41 ± 0.42 | 74.53 ± 1.68 | 59.37 ± 0.87 | 60.25 ± 0.33 |
| A3:A4 | 74.94 ± 0.38 | 68.72 ± 1.73 | 79.19 ± 1.75 | 65.19 ± 11.69 | 99.66 ± 0.26 | 76.98 ± 1.99 | 60.62 ± 1.12 | 60.25 ± 0.34 |
| A3:A5 | 74.16 ± 1.95 | 68.32 ± 0.75 | 78.77 ± 2.19 | 67.90 ± 11.28 | 99.79 ± 0.14 | 75.17 ± 1.14 | 59.65 ± 0.74 | 60.24 ± 0.41 |
| A4:A4 | 75.26 ± 1.45 | 69.74 ± 0.62 | 78.52 ± 2.35 | 64.20 ± 11.01 | 99.69 ± 0.28 | 76.86 ± 0.97 | 59.01 ± 0.97 | 59.71 ± 0.38 |
| A4:A5 | 74.50 ± 1.36 | 68.11 ± 1.00 | 79.24 ± 1.69 | 65.93 ± 11.67 | 99.78 ± 0.03 | 75.07 ± 1.12 | 59.45 ± 0.82 | 59.78 ± 0.24 |
| A5:A5 | 73.33 ± 1.19 | 66.69 ± 1.22 | 77.75 ± 1.45 | 65.93 ± 10.75 | 99.83 ± 0.16 | 74.67 ± 0.73 | 59.24 ± 0.67 | 59.28 ± 0.42 |

## G Complexity Analysis

Last, we discuss the efficiency of AdT-HyGCL in terms of space complexity and time complexity. *Space complexity*: Since we adopt a sparse representation to store $H$, the space should be $\mathcal{O}(E)$, where $E = \sum_e d(e)$. The size of node feature $\mathcal{X}$ is $\mathcal{O}(Nd)$, where $d$ is the dimension of node features. Therefore, the space complexity of storing an augmented hypergraph pair is $\mathcal{O}(2E+2Nd)$. When calculating the contrastive loss over large-scale datasets, we leverage mini-batch data which takes $\mathcal{O}(N'^2)$, where $N'$ is the batch size and $N' \ll N$. Note that this similarity matrix is shared to dynamically adjust the temperature index. Moreover, the space complexity of storing the parameter matrix of backbone AllDeepSets is $\mathcal{O}(Nh+Mh+2hb)$, where $h$ is the hidden dimension and $b$ is the output embedding dimension. Therefore, the space complexity of AdT-HyGCL is $\mathcal{O}(2E + 2Nd + N'^2 + Nh + Mh + 2hb)$, which is linear to the size of nodes. *Time complexity:* The time complexity for updating the temperature index $\tau^{(t)}$ for each epoch is $\mathcal{O}(N'^2)$ per each view, where $N' \ll N$. Since we have a local view and a global view strategy, the total time complexity for this part is $\mathcal{O}(2N'^2)$. When computing full negatives, the time complexity for calculating the contrastive pairs is linear to the size of nodes, i.e. $\mathcal{O}(N)$. So the total time complexity for each round is $\mathcal{O}(2N'^2 + N)$, again showing its efficiency.

