# OpenReview forum: "Adaptive Temperature Enhanced Dual-level Hypergraph Contrastive Learning"
_ICLR.cc/2024/Conference — Submitted to ICLR 2024_

### Official Review · Reviewer_cbYE · 2023-10-29

**Soundness:** 1 poor
**Presentation:** 1 poor
**Contribution:** 2 fair
**Rating:** 1
**Confidence:** 5

**Summary:**

In this paper, the authors design AdT-HyGCL, an adaptive temperature-enhanced dual-level hypergraph contrastive learning model. It overcomes the limitations of existing hypergraph contrastive learning by capturing both individual and group-wise behaviors. The adaptive temperature-enhanced contrast optimization improves discrimination, enhancing hypergraph representation learning. Theoretical justifications and empirical experiments on eight benchmark hypergraphs demonstrate AdT-HyGCL's effectiveness and robustness.

**Strengths:**

S1: The studied issue of hypergraph contrastive learning is important.

S2: The baseline methods selected for this study are well-suited for comparison and evaluation.

**Weaknesses:**

W1: The presentation of the paper requires to be improved heavily. Some examples are as follows:

(i) In the listed contributions, the statements of "Novelty", "Generalization", and "Effectiveness and Robustness" are improper. The authors should state the specific contributions of this work here, e.g., "Novelty"->"A novel adaptive temperature enhanced
hypergraph contrastive learning framework".

(ii) In Preliminaries, the statement "The main idea of graph contrastive learning aims to maximize the agreements among positive and negative contrastive pairs over" is not clear. Using a more clear statement "The main idea of graph contrastive learning aims to maximize the similarity between positive pairs and meanwhile minimize the similarity between negative pairs" is better. Also, in Equation (2), sim($\cdot$) is not given a definition or explanation.

(iii) In Methodology, the graph augmentation process itself adds noise to the original graph, so the module with the name "noise-enhanced hypergraph augmentation" is strange. It may be better if the authors replace "noise-enhanced" with "feature distortion based".  The symbols of $x_{i}$ and $\delta_{i}$ should be bolded as they represent vectors, not values. The definition of the concatenation operator is ambiguous: sometimes they are denoted as $[\cdot:\cdot]$ and sometimes they are denoted as $\oplus$. In Equation (5), $\eta$ has unclear statements and it represents the learning rate of the temperature adjustment.

(iv) Many contents in the main body are not self-contained, e.g., Table 3 should be moved into the main body.


W2: Lacking necessary analysis of the important loss function. The authors do not analyze the JSD loss in the paper, but why they can conclude that  AdT-HyGCL is suitable for JSD loss (note that JSD loss is totally different from NT-Xent loss)?

W3: The presented experimental results have lots of errors. Some examples are as follows:

(i) In Table 1, for Macro-F1 results, the best result should be "64.98 ± 1.77" and the runner-up should be "64.65 ± 0.85" on Citeseer.

(ii) In Table 2, the best result should be "59.65 ± 1.62" on House (Nettack).

These errors make the whole experimental analysis not convincing.

W4: Lacking some important baseline comparison. The paper does not compare AdT-HyGCL with other temperature adjustment methods (e.g., the methods in [1], [2], and [3]), which limits the ability to assess the effectiveness of AdT-HyGCL relative to other methods.

[1] Chen, Jiawei, et al. "Adap-$\tau$: Adaptively Modulating Embedding Magnitude for Recommendation." WWW2023.

[2] Qiu, Zi-Hao, et al. "Not All Semantics are Created Equal: Contrastive Self-supervised Learning with AutomaticTemperature Individualization." ICML2023.

[3] Zhang, Oliver, et al. "Temperature as uncertainty in contrastive learning." arXiv preprint arXiv:2110.04403(2021).

**Questions:**

Please see my previous comment.

---

### Official Review · Reviewer_xcsf · 2023-10-30

**Soundness:** 2 fair
**Presentation:** 2 fair
**Contribution:** 2 fair
**Rating:** 3
**Confidence:** 5

**Summary:**

An adaptive temperature-enhanced dual-level hypergraph contrastive learning model (called AdT-HyGCL) is introduced to enhance hypergraph contrastive learning. AdT-HyGCL designs a dual-level contrast mechanism to capture both individual behaviors and group-wise behaviors. Also, it introduces adaptive temperature-enhanced contrast optimization to improve discrimination ability between contrastive pairs. The experimental results on several hypergraph datasets demonstrate the effectiveness of AdT-HyGCL.

**Strengths:**

1. This paper designs a hypergraph contrastive learning framework that integrates the dual-level contrast strategy and the adaptive temperature-enhanced contrast optimization to pre-train the HyGNNs encoder.

2. AdT-HyGCL uses several hypergraph augmentations and contrast optimizations to improve the hypergraph representation learning. The experimental results on several hypergraph datasets demonstrate the effectiveness of AdT-HyGCL.

3. The paper is well organized.

**Weaknesses:**

1. The motivation of this work is not good enough.
For example, the authors claim that "they (graphs) have limitations in captureing ... higher-order group-wise structures.'' What is the meaning of high-order group-wise structures in this paper? Why is the statement correct? Considering the higher-order structures (such as communities) are widely studied in the field of graph data analysis, it is better to showcase the viewpoints carefully.
Furthermore, the authors claim that "current contrastive learning methods over hypergraphs still have limitations in modeling the high-order relationships ... within hypergraphs ...". Then, maybe hyper-hypergraphs, if any, cannot capture the high-order relationshi within hyper-hypergraphs. Is it really interesting? Should we propose another "dual-level contrast mechanism" for the hyper-hypergraphs?


2. The novelty of this study is not clear.

(1) The novelty of the concept of `noise-enhanced' should be clearly presented. The stategy of performing noises to node attribute features has been well utilized in the literature, such as [1]. The authors should carefully discussed them in the manuscript, especially when [1] has been cited as [Zhu, et al. 2021b] in the paper.

(2) The idea of using community structures in constrastive learning has been proposed in the literature, such as [2]. The authors should clearly present this point. In addition, it seems that the ``community'' in this study does not match the well-known community in the literature. Then, it is better to use another phrase in this work to aoivd the ambiguiity.

(3) The idea of the adaptive temperature-enhanced contrast optimization in pre-tained models (including GNNs) has been proposed in the literature, such as [3-6]. The authors should clearly discuss the differences between this work and the literature.

3. The methods of the paper should be carefully presented.
Considering the adaptive temperature-enhanced contrast optimization in pre-tained models (including GNNs) has been proposed in the literature, such as [3-6], the authors should clearly present the relationship between the propostions as well as their proofs and the literature. In addition, the proofs of these propositions should be carefully presented rather than giving one or more examples. For instance, it is stated that ``the temperature descents'' in the paper. Why must the temperature descent during the training process for any loss? The authors should prove it seriously.

4. The experiments of the paper should be improved.
As shown in Table 1 and Table 2, the experimental results of the proposed AdT-HyGCL are not always the best. Maybe the authors should analyze and present the reasons. Also, the parameter sensitivity experiments should be provided, such as $\tau_{low}$, $\eta$, $\rho$, $\lambda_1$ and $\lambda_2$. In addition, more recent studies on hypergraph contrastive
learning shoud be considered as baselines, such as [7][8].

[1] Graph Contrastive Learning with Adaptive Augmentation. Y. Zhu, Y. Xu, F. Yu, Q. Liu, S. Wu, L. Wang. WWW 2021.

[2] Graph Communal Contrastive Learning. B. Li, B. Jing, H. Tong. WWW 2022.

[3] Temperature as Uncertainty in Contrastive Learning. O. Zhang, M. Wu, J. Bayrooti, N. Goodman. arXiv:2110.04403. 2021.

[4] Understanding the Behaviour of Contrastive Loss. F. Wang, H. Liu. CVPR. 2021.

[5] Rethinking Temperature in Graph Contrastive Learning. Z. Liu, H. Feng, C. Wang. 2021.

[6] Temperature Schedules for Self-Supervised Contrastive Methods on Long-Tail Data. A. Kukleva, M. Bohle, B. Schiele, H. Kuehne, C. Rupprecht. ICLR 2023.

[7] Cross-view graph contrastive learning with hypergraph. J. Zhu, W. Zeng, J. Zhang, J. Tang, X. Zhao. Information Fusion. 2023.

[8] Collaborative contrastive learning for hypergraph node classification. H. Wu, N. Li, J. Zhang, S. Chen, M. K. Ng, J. Long. Pattern Recognition. 2023.

**Questions:**

See the points in the Weaknesses.

---

### Official Review · Reviewer_r4B1 · 2023-10-31

**Soundness:** 2 fair
**Presentation:** 3 good
**Contribution:** 1 poor
**Rating:** 3
**Confidence:** 5

**Summary:**

This paper proposes an adaptive temperature-enhanced hypergraph learning framework, AdT-HyGCL, for self-supervised hypergraph learning, which integrates the dual-level contrast strategy and the adaptive temperature-enhanced contrast optimization to improve hypergraph contrastive learning. Theoretical justifications and empirical experiments justify the effectiveness of AdT-HyGCL.

**Strengths:**

1. This paper studies an important problem of self-supervised hypergraph learning.
2. The paper is well-organized and clearly written.

**Weaknesses:**

1. The novelty of the proposed framework in this paper is incremental. Both multi-level contrast and adaptive temperature are not novel ideas. The former has been explored in many previous works [1-3], while the latter has been studied in [4]. The authors didn't cite these works, which should be referred to, discussed, and compared.
2. The performance improvement seems limited in Table 1.
3. Table 2 misses several competing baselines in Table 1, like CHGNN.

[1] Duan, Jingcan, et al. "Graph anomaly detection via multi-scale contrastive learning networks with augmented view." AAAI 2023.

[2] Ju, Wei, et al. "Unsupervised graph-level representation learning with hierarchical contrasts." Neural Networks 2023.

[3] Liu, Yanbei, et al. "Multi-Scale Subgraph Contrastive Learning." IJCAI 2023.

[4] Zhang, Hongjun, et al. "Sleeppriorcl: Contrastive representation learning with prior knowledge-based positive mining and adaptive temperature for sleep staging." arXiv preprint arXiv:2110.09966 (2021).

**Questions:**

See above.

---

### Official Review · Reviewer_1PBv · 2023-10-31

**Soundness:** 3 good
**Presentation:** 3 good
**Contribution:** 2 fair
**Rating:** 5
**Confidence:** 3

**Summary:**

This paper focused on hypergraph contrastive learning. Previous papers only deal with the node level information instead of the group (community) information. The paper also applied adaptive temperature technique when discriminating contrastive pairs. This allows the model to adaptively optimize the positive and negative pairs.

The authors first introduced a noise enhanced hypergraph augmentation, which includes 5 different ways. Then a two stage contrastive learning is proposed. It includes the node-level and community-level.

The experiments show that the paper outperforms other hypergraph models and hypergraph contrastive learning models in accuracy, macro-F1, robustness. The ablation study shows that different hypergraph augmentation methods can help increase the performance. Among all augmentation methods, hyperedge removing is the most important one. The temperature analysis indicates that the adaptive temperature is helpful.

**Strengths:**

This paper propose a contrastive learning method for hypergraph, and outperforms the baselines including both hypergraph algorithms and hypergraph contrastive learning algorithms. The paper is well written and easy to follow. The proposed algorithm is simple yet effective. The ablation study shows which augmentation algorithm is helpful. It turns out hyperedge removing is the most helpful augmentation, which is intuitive.

**Weaknesses:**

Typo:, in Figure 1, Dual-Leve -> Dual-Level Hypergraph Contrastive Learning.

For the citeseer's Macro F1, it seems CHGNN performs slightly better.

And some common datasets like DBLP and Pubmed seems missing in the paper.

For House and walmart, most baseline models performs poorly, probably due to the heterophily intrinsic.

Only one data augmentation method is new, which is HyedgeR. I'm a little concern about the novelty since it's intuitive.

**Questions:**

Overall, the performance greatly outperform the hypergraph models, while slightly outperform the contrastive learning models. So this work seems focus on contrastive learning instead of hypergraph models. However, i'm not sure whether this is novel in contrastive learning. More details regarding to this in the related work may be appreciated.

Meanwhile, the $\tau_{low}$ is a hyperparameter. Without the $\tau_{low}$ the model will have a significant performance decrease. I'm curious how to determine the $\tau_{low}$. Is there any method? Or it is decided based on empirical result.

---

### Official Review · Reviewer_AN7j · 2023-11-01

**Soundness:** 3 good
**Presentation:** 3 good
**Contribution:** 3 good
**Rating:** 5
**Confidence:** 3

**Summary:**

This paper introduces a dual-level hypergraph contrastive learning framework, AdT-HyGCL. It incorporates a dual-level contrast mechanism (the node level and community level) and an adaptive temperature-enhanced approach to improve the effectiveness of contrastive learning over hypergraphs. The research addresses certain limitations observed in existing works focusing on modeling high-order relationships over unlabeled data. Extensive experiments show the effectiveness of the proposed AdT-HyGCL.

**Strengths:**

The novelty of this paper lies in its endeavor to overcome the constraints encountered in prior research pertaining to the modeling of high-order relationships within unlabeled data. It shifts its emphasis towards the intricate realm of group-wise collective behaviors within hypergraphs. Additionally, it introduces an innovative adaptive temperature coefficient designed to autonomously fine-tune the representation distances. As for clarity, I think this paper is written in a lucid and comprehensible manner, ensuring ease of understanding for its readers. Theoretical justifications meet the requirements, and empirical experiments are abundant.  The authors demonstrated through a substantial number of experiments that the method exhibits advantages in terms of generality, effectiveness, and robustness. This method has achieved SOTA performance in hypergraph contrastive learning, which I believe holds implications.

**Weaknesses:**

The methodology section of this paper appears to be somewhat limited, particularly in section 4.2.2, which lacks a comprehensive theoretical explanation of 'COMMUNITY-LEVEL HYPERGRAPH CONTRASTIVE LEARNING'. Additionally, it may be beneficial to provide an exposition on how these two branches are integrated to obtain the final representations.

**Questions:**

1. I have some difficulty in comprehending how the adaptive temperature parameter is implemented. Could you provide more formulaic evidence or details of its code implementation?
2. Why were UniGCNII and AllDeepSets chosen for the robustness experiments instead of a seemingly more effective method like CHGNN? Can you provide the robustness experiment results for CHGNN?
3. I am curious about how the final representations obtained from the "node-level" and "community-level" branches in the paper are combined. On a theoretical level, how is the balance between the results obtained from these two branches achieved?

---

### Meta-Review · Area_Chair_bkTs · 2023-11-30

**Metareview:**

**Summary:** The paper introduces AdT-HyGCL, a novel dual-level hypergraph contrastive learning framework that addresses limitations in existing works by considering both node-level and community-level information.  It incorporates a dual-level contrast mechanism and an adaptive temperature-enhanced approach to improve hypergraph contrastive learning.  The experimental results shows the superiority of AdT-HyGCL over other selected hypergraph models and provide insights into the importance of different hypergraph augmentation methods.  The adaptive temperature analysis further supports its effectiveness.  However, the paper has several weaknesses identified by the reviewers, including the need for significant improvements in presentation, errors in experimental results, lack of analysis on the JSR loss function, unclear novelty and differentiation, and the omission of comprehensive sensitivity experiments with common datasets.

The authors did not provide a rebuttal to the reviewers concerns.  Overall, I see an agreement towards the rejection of the paper due to the limitations raised by the reviewers.

**Strengths:** The paper addresses the constraints of prior research in modeling high-order relationships in unlabeled data through its focus on group-wise collective behaviors in hypergraphs.  It introduces an adaptive temperature coefficient for representation distance fine-tuning. The paper is well-written and clear, with theoretical justifications and abundant empirical experiments.  The proposed method outperforms baselines in hypergraph contrastive learning, achieving state-of-the-art performance.

**Weaknesses:**  The reviewers raised several concerns regarding the paper. They highlighted the need for significant improvement in the paper's presentation in several areas of the paper.  Additionally, errors in the presented experimental results were identified, leading to an unconvincing analysis.  The lack of analysis on the crucial JSR loss function and a comparison with other temperature adjustment methods was also noted. The reviewers expressed doubts about the novelty of the study, calling for clearer articulation of the differences between this work and existing literature.  Moreover, they pointed out that the method of adaptive temperature-enhanced contrast optimization had already been presented in other papers.  The paper was deemed to lack a clear demonstration of the need for the study, raising questions about its motivation. Furthermore, improvements in the experimental results, including more comprehensive sensitivity experiments with common datasets, were suggested, along with the need to address spelling errors.

**Justification For Why Not Higher Score:**

The paper received significant criticism from the reviewers, who raised several substantial flaws that seem challenging to address. It appears evident that the paper is not yet suitable for publication. Additionally, the authors did not make an effort to explain or challenge the raised concerns from the reviewers.

**Justification For Why Not Lower Score:**

N/A

---

### Decision · Program_Chairs · 2024-01-16

Reject